# Effective Method for the Determination of the Unit Cell Parameters of New MXenes

**DOI:** 10.3390/ma15248798

**Published:** 2022-12-09

**Authors:** Alexander Syuy, Dmitry Shtarev, Alexey Lembikov, Mikhail Gurin, Ruslan Kevorkyants, Gleb Tselikov, Aleksey Arsenin, Valentyn Volkov

**Affiliations:** 1Institute of High Technologies and Advanced Materials, Far Eastern Federal University, 690922 Vladivostok, Russia; 2Center for Photonics and 2D Materials, Moscow Institute of Physics and Technology, 141701 Dolgoprudny, Russia; 3Hong Kong Quantum AI Lab Ltd., Hong Kong Science and Technology Parks Corporation, Hong Kong, China

**Keywords:** MAX phase, MXenes, unit cell, crystal lattice, 2D materials, multilayered materials

## Abstract

MXenes are of great practical interest. While the physical properties of such a well-known MAX phase as Ti_3_AlC_2_ and the Ti_3_C_2_ MXene that is based on it have been widely studied, it is extremely important to study the properties of new four-component MAX-phases and the MXenes based on them. To do this, first, it is necessary to characterize the obtained materials. In this work, the Ti_3−x_Nb_x_C_2_ MXene was characterized. Since the material is fairly new, there are no crystallographic data for such systems in the international databases. We proposed a method for the determination of the main unit cell parameters of the new Ti_3−x_Nb_x_C_2_ MXene, which was based on a combination of the DFT method, TEM studies, and an X-ray diffraction analysis.

## 1. Introduction

Since Ti_3_C_2_ was first reported in 2011 [1], the family of two-dimensional (2D) transition metal carbides and nitrides (MXenes) has expanded significantly and has influenced such fields as energy storage [2], electromagnetic interference shielding [2], optoelectronics [3], water desalination [3], catalysis, medicine, and many others [4]. MXenes have the general chemical formula M_n+1_X_n_T_x_, where M represents an early transition metal, X is a carbon or nitrogen, n = 1–4, and T denotes the surface termination group [5]. MXenes are usually synthesized through selective chemical etching methods; in particular, group-A atomic layers are removed from their stacked MAX phase precursors [6]. The relatively weak M–A bond in the MAX phase is metallic and chemically active and, thus, is easier to break than the mixed covalent–ionic M–X bonds [7]. Therefore, M–A bonds can be selectively broken by etchants during the formation of MX layers (MXenes) [8]. A unique feature of MXenes manifests itself when two transition metals are mixed in a MXene structure [2]. In addition to the formation of the expected solid solutions, such as (Ti,Nb)CT_x_, transition metals can form ordered structures in a single 2D MXene flake either by forming atomic sandwiches of transition metals planes (for n ≥ 2), such as Mo_2_TiC_2_T_x_, or in-plane (n = 1) ordered structures, such as (Mo_2/3_Y_1/3_)_2_CT_x_ [9].

The rapid development of new ordered 2D carbide phases has caused a rush in the MAX phase research community [10]. To date, more than 150 MAX phases formed from a combination of 14 M and 16 A elements have been reported [11]. Since 2017, researchers have synthesized about 30 new ordered MAX metal phases with double-transition-metal junctions and have studied their properties, including their magnetic [12,13,14], mechanical [15,16], electronic [17], and thermal characteristics [18]. The formation of solid solutions on M and/or X sites opens the possibility of synthesizing an infinite number of nonstoichiometric MXenes and the attractive possibility of fine-tuning the properties by mixing different transition metals or by creating carbonitrides [19]. Thus, recently, researchers have demonstrated the superior electrochemical properties of Ti/Nb based MAX and MX phases [20]. The electronic properties and electrochemical performance of the double-metal MXenes TiNbC and TiNbCT_2_ (T = O, F, and OH) as anode materials in Li-ion batteries (LIBs) have been studied to understand the effects of Nb(Ti)-layer inserting in the Ti_2_C (Nb_2_C) monolayer and in terminated functional groups (T=O, F, and OH) on the electrochemical performance [20]. It should be noted that the electronic, optical, chemical, and catalytic properties of ternary MAX and MX compounds can be predicted using the density functional method (DFT) [8]. To make these predictions more reliable, one should, a priori, know the crystallographic structure of the studied compound. X-ray diffraction (XRD) is one of the most powerful tools for the direct determination of the crystallographic structure of MXenes [21]. Since MAX phases are fairly new materials and since there is a lack of information in the international databases on the interpretation of X-ray transcripts, the characterization of new compositions of MXenes and MAX phases is complicated. In this article, we presented a new approach for the refinement of the structure of MXenes, which had a wide range of applications. It consisted in the modification of the crystallographic information file (cif) of the parent MAX phase. This approach was successfully applied to MXenes with a mixed-cation composition (Ti_3−x_Nb_x_C), while the structure of the 312th MAX phase (Ti_3_AlC_2_) was taken as the basis. The new approach made it possible not only to decode the experimental X-ray diffraction patterns but also to perform a DFT simulation of the MXenes. Previously, this approach was not used to refine the elementary cell. This approach could claim to be universal in the determination of the unit cell parameters, including the positions of alloying element atoms, of MXenes.

## 2. Materials and Methods

MXene Ti_3−x_Nb_x_C_2_ was purchased from Advanced 2D Materials Co., Ltd., Shanghai, China. Lateral size: 1–5 μm. Product performance: high purity and excellent electrochemical performance. Functional group: –OH, –F, –O, and –Cl (customizable). Purity: 99%. Storage conditions: dry at room temperature. Ti_3−x_Nb_x_C_2_ MXene was obtained by etching Al from the (Ti,Nb)_3_AlC_2_ MAX phase with a mixture of HCl and LiF acids.

The powder diffraction data for Rietveld analysis was collected at room temperature using Bruker D8 ADVANCE powder diffractometer (Cu-Kα radiation) and linear VANTEC detector. The step size of 2θ was 0.016°, and the counting time was 1.5 s per step. Rietveld refinement was performed using TOPAS 4.2 [17]. Crystal lattice expansion as a function of temperature was studied using the low-temperature chamber TTK 600 (Anton Paar).

The sample morphology was determined through scanning electron microscopy (FEI Scios 2, Thermo Scientific, Waltham, MA, USA). The composition of chemical elements and their distribution in the synthesized heterostructures were established through energy-dispersive X-ray spectroscopy (EDAX, Octane Elite, AMETEK, Berwyn, PA, USA).

Images of Ti_3−x_Nb_x_C_2_ MXenes were obtained using high-resolution transmission electron microscopy (TEM, JEOL JEM-2100 microscope, Japan) equipped with 200 kV field emission gun and point resolution of 0.19 nm. For TEM measurements, Ti_3−x_Nb_x_C_2_ MXene samples were mixed in isopropanol and were applied to a TED PELLA, Inc. (Redding, CA, USA), copper grid (Lacey Carbon support films, 200 mesh) until completely dry. TEM-EDX studies were carried out using Aztech X-Max 100 analytical attachment for energy dispersive analysis.

The electronic structures of MXenes were calculated using periodic DFT approach. For this purpose, GGA Perdew–Burke–Ernzerhof (PBE) density functional [22,23] as implemented in the ABINIT 6.8.3 program [23] was chosen. The basis set in the form of Troullier–Martins norm-conserving pseudopotentials [24] with the kinetic energy cut-off of 30 Hartree was employed. The Brillouin zone (BZ) was sampled from the automatically generated Γ-point-centered 7 × 7 × 3 Monkhorst–Pack grid of *k*-points [25]. The default total energy convergence criterion of 1.0 ×10^−8^ Hartree was applied. Band structures and density of states were plotted using the Gnuplot 5.2 software package [26].

## 3. Results

### 3.1. SEM and TEM Characterizations

Figure 1a demonstrates the typical particle of Ti_3−x_Nb_x_C_2_. It shows that the MXene particles had a cubic shape and reached several microns in size. They may form agglomerates of a few dozens of microns. The corresponding EDX-mapping is presented in Figure 1b. One can see that the main constituting chemical elements Ti, Nb, and C were distributed evenly over the sample surface (in Figure 1b, the carbon signal was visible only on the substrate, which was due to the presence in the sample of a large amount of niobium, which gave a strong signal). In addition to that, a small amount of the parent MAX-phase was visible close to the Al sites. The fragment in Figure 1a that is enclosed in a dashed rectangle is depicted and enlarged in Figure 1c. One can see that the cubic particle was comprised of nanosheets of ~80–120 nm in thickness. The EDX-spectrum of the particle showed (Figure 1d) a significant amount of Nb—the synthesized compound could be described as Ti_1.625_Nb_3.375_C_4_.

Figure 2 shows a TEM image, electron diffraction pattern, and EDX analysis of the Ti_3−x_Nb_x_C_2_ particle. Figure 2b demonstrates the multilayer structure of the MXene corresponding to the SEM image in Figure 1c. The electron diffraction patterns (insets in Figure 2b) showed bright spots related to the Ti_3−x_Nb_x_C_2_ nanocrystals with a certain orientation of the atomic planes. The presence of a dot diffraction pattern against the background of an annular diffraction pattern most likely indicated that a significant part of the nanocrystals of the studied MXene had the same spatial orientation. The high-resolution TEM image (insets in Figure 2b) showed that the distance between the atomic planes equaled 0.21 nm.

The TEM-EDX spectrum (Figure 2c) showed peaks corresponding to the MXene material (Ti, Nb, and C), O (due to surface oxidation), F, Cl (due to the etching of the MAX phase in a mixture of HCl and LiF), Al (remained after the etching of the MAX phase), and Cu (due to the copper grid that was used). The sample composition established from the EDX spectrum (Ti_2.15_Nb_3.85_C_4_) was close to that derived from the SEM (Ti_1.625_Nb_3.375_C_4_), which let us conclude that the actual ratio of Ti:Nb was ~1:2. Thus, the sample composition may correspond to the formulae Ti_2_Nb_4_C_4_.

### 3.2. X-ray Diffraction

Figure 3 shows the X-ray diffraction pattern of the Ti_3−x_Nb_x_C_2_ sample. Its structure determination was based on the structure # 312 of MAX Ti_3_AlC_2_ (cif #7221324 in [27,28,29,30,31,32,33]), which was represented by diffractogram 1 in Appendix A. It had a hexagonal crystal lattice P6_3_/mmc (a = b = 3.072 Å, c = 18.73 Å, α = β = 90°, γ = 120°). Appendix A shows that the X-ray diffraction pattern of the parent MAX-phase did not match that of the analyzed MXene. Thereafter, the Al atoms were eliminated from the above Ti_3_AlC_2_ structure followed by the simulation of the X-ray diffraction pattern of the remaining MXene Ti_3_C_2_. The obtained X-ray diffraction pattern 2 in Appendix A showed that the elimination of the Al was accompanied with a change in the relative intensities of most of the reflections that was related to the disappearance of certain planes that were diffracting X-rays. This was especially noticeable in the example of plane (104) (when comparing the relative intensities of curves 1 and 2 in Appendix A in a region of about 40 degrees). In this plane exactly, the excluded aluminum atoms were located, which can be clearly seen in Appendix A. However, the structure of MXene Ti_3_C_2_ obtained through the simple elimination of the Al atoms in Ti_3_AlC_2_ differed from its actual structure due to the possible shrinking of the crystal lattice along the **c**-axis.

To account for the shrinking phenomenon, one should consider the reflection at ~12° of experimental MXene X-ray diffraction pattern 4 in Appendix A and the reflection at ~9° of parent MAX-phase X-ray diffraction patterns 1 and 2 in Appendix A. This reflection was due to the plane (002) depicted in Appendix A corresponding to the interplane distances of 9.436 Å (the parent MAX-phase) and 6.896 Å (the analyzed MXene).

By decreasing the lattice parameter **c** of the parent MAX-phase without Al atoms while leaving the other lattice parameters unchanged, one can shift the characteristic reflection due to the plane (002) moving towards the experimentally observed value (X-ray diffraction pattern 3 in Appendix A). This was achieved using the crystal lattice parameters a = b = 3.079 Å, c = 13.792 Å, α = β = 90°, and γ = 120° from which the lattice parameters a = b = 3.053 Å, c = 13.7213 Å, α = β = 90°, and γ = 120° were obtained. The lattice parameters were refined through the processing of the experimental diffraction data using the Le Bail fitting method (Appendix A). The comparison of X-ray diffraction patterns 3 and 4 in Appendix A revealed a good match with most of the experimentally observed and simulated reflections.

The reflection intensities in Appendix A deserved particular attention. X-ray diffraction pattern 1 in Appendix A showed that the most intense reflection of the parent MAX-phase observed at ~38.9° was due to the crystallographic plane (104). This plane contained two Ti and two Al atoms (Figure 4a). Upon elimination of its Al atoms, this reflection was no longer the most intense (X-ray diffraction pattern 2 in Appendix A) since the structure contained no Al atoms. The most intense reflection of experimental X-ray diffraction pattern 4 in Appendix A was also observed at ~38.9°. However, it was due to plane (103) (Figure 4b). The corresponding interplane distance of 2.306 Å was very close to 0.21 nm—the interplane distance determined earlier using the high-resolution TEM method (see the inset in Figure 2). Generally, it should be noted that the data obtained using TEM and XRD matched well. Both data agreed well with the simulated X-ray diffraction pattern that was based on the structure of the parent MAX-phase without Al atoms (Appendix A).

The comparison of the simulated (red) and experimental (black) MXene X-ray diffraction patterns on the left of Figure 5 showed that some reflections of the latter had anomalously high intensities (Figure 5a). This could be due to the fact that, in the structure of the actual MXene Ti_3−x_Nb_x_C_2_, some Ti atoms were substituted by those of Nb, featuring a stronger ability to reflect X-rays. If so, then Nb atoms should replace those of Ti that were exactly at the positions of those laying in the planes depicted in the Figure 5b.

To verify the above hypothesis, we performed DFT modeling on the mixed-cation MXene Ti_3−x_Nb_x_C_2_. As was demonstrated earlier, the SEM-EDX and TEM-EDX data predicted the Ti_2_Nb_4_C_4_ composition of the synthesized MXene. In total, this composition featured fifteen unique combinations of the two Ti and four Nb atoms in the cationic sublattice. They could be grouped according to the energy and spatial symmetry of the unit cell as shown in Table 1.

The conducted computations showed that the most energetically favorable structure and the only one matching the experimental lattice symmetry structure of Ti_2_Nb_4_C_4_ contained Ti atoms in the corner and at the edge of the unit cell while all the Nb atoms were residing in its volume.

The performed DFT-modeling made it possible to calculate the XRD pattern of the MXene with titanium atoms partially substituted for niobium atoms (Appendix A). It could be seen that this substitution led to a decrease in the relative intensity of the reflection in the region of 12 degrees and an increase in the intensity of the peaks in the region of 30–45 degrees, which fully corresponded to the experimentally observed pattern.

## 4. Conclusions

The conducted experimental and theoretical study of the mixed-cation MXene Ti_3−x_Nb_x_C_2_ enabled the determination of its main crystal lattice parameters. It was found that Nb atoms occupied the positions of the Ti atoms in the structure of Ti_3_C_2_. The experimental SEM-EDX and TEM-EDX data suggested the Ti_2_Nb_4_C_4_ composition of the synthesized MXene. The performed DFT modeling, in turn, predicted that the lowest energy structure of Ti_2_Nb_4_C_4_ contained Ti atoms in the corner and at the edge of the compound’s unit cell. The corresponding unit cell symmetry perfectly matched the experimentally determined one.

## Figures and Tables

**Figure 1 materials-15-08798-f001:**
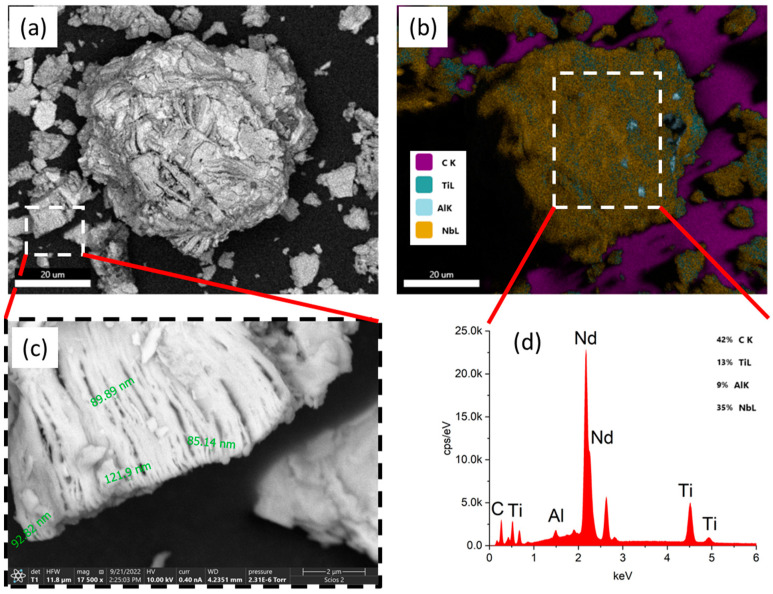
(**a**) SEM images of typical Ti_3−x_Nb_x_C_2_ particle; (**b**) EDX surface mapping of the particle; (**c**) enlarged fragment of cubic Ti_3−x_Nb_x_C_2_ particle; (**d**) EDX spectrum of a characteristic Ti_3−x_Nb_x_C_2_ particle.

**Figure 2 materials-15-08798-f002:**
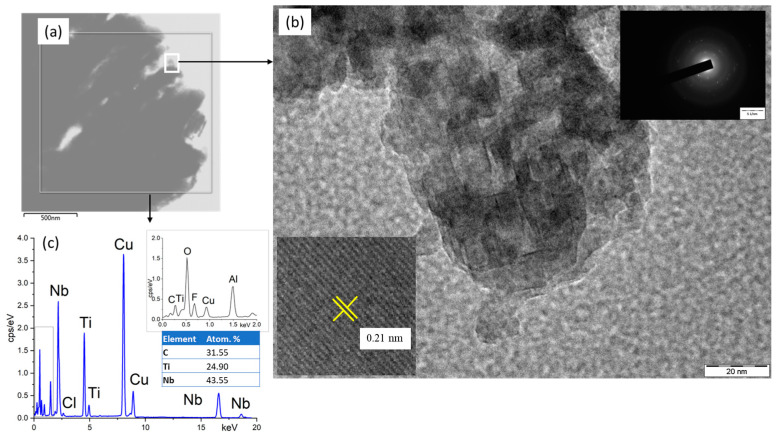
TEM image of typical Ti_3−x_Nb_x_C_2_ particle. (**a**) Macrophotography; (**b**) areas studied through electron diffraction; (**c**) elemental composition from EDX.

**Figure 3 materials-15-08798-f003:**
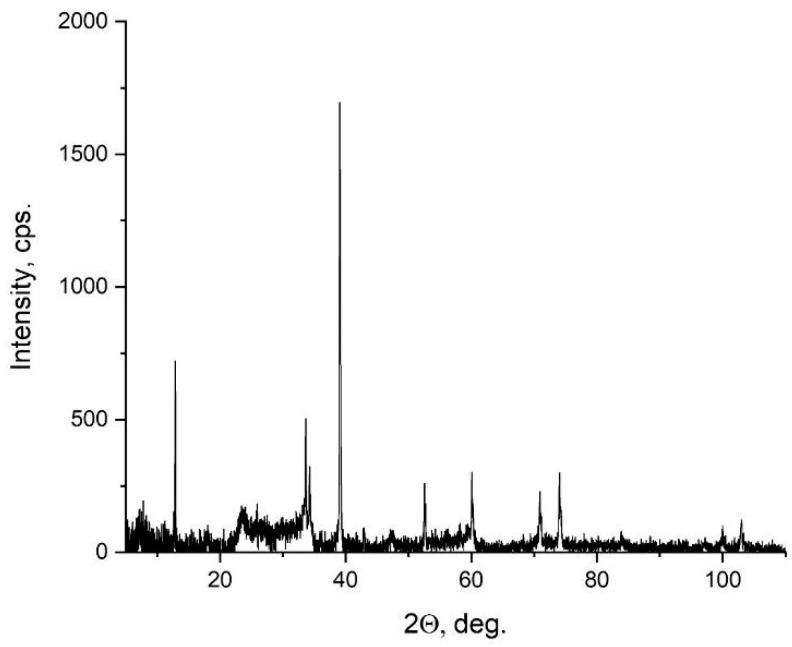
X-ray diffraction pattern of the studied structure of the Ti_3−x_Nb_x_C_2_ sample.

**Figure 4 materials-15-08798-f004:**
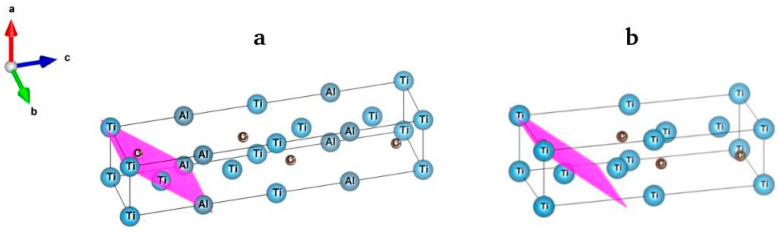
(**a**) Crystallographic plane (104) in Ti_3_AlC_2_; (**b**) crystallographic plane (103) in the analyzed MXene.

**Figure 5 materials-15-08798-f005:**
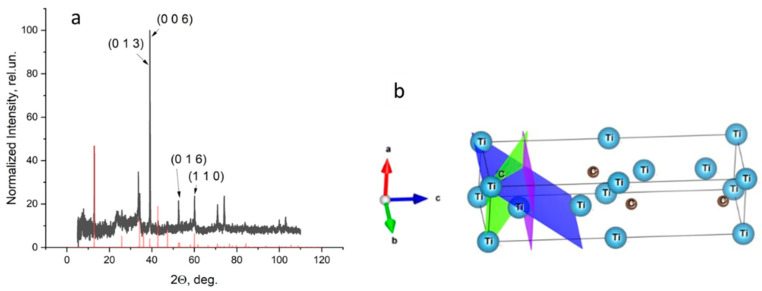
Experimental (black line) and calculated (red columns) X-ray patterns of the studied MXene (**a**,**b**) elementary lattice with crystallographic planes labeled in (**a**).

**Table 1 materials-15-08798-t001:** Relative energies and symmetries of the Ti_2_Nb_4_C_4_ MXenes.

Energy, eV	Symmetry	Ti Atom Positions According to Appendix A
0.00	P6_3_/mmc (#194)	1–2
2.56	P3m1 (#156)	1–5, 1–6, 2–3, 2–4
3.20	P3m1 (#156)	1–3, 1–4, 2–5, 2–6
5.12	P-6m2 (#187)	3–6, 4–5
5.13	P6_3_mc (#186)	3–5, 4–6
5.22	P-3m1 (#164)	3–4, 5–6

## Data Availability

Not applicable.

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
