# Peer review of "Effective Method for the Determination of the Unit Cell Parameters of New MXenes"

_materials, 2022, doi:10.3390/ma15248798_

Round 1

Reviewer 1 Report

I am grateful for this opportunity to review the manuscript by Alexander Syuy et al submitted to MDPI Materials. The research problem of solving the structure of novel inorganic materials, such as quaternary MAX and ternary MXene systems chosen for this study, is, indeed, a very important one. The method the authors propose to solve the structure of a mixed TiNbC is based on combining the structural information retrieved from diffraction studies with the theoretical modeling of cation distribution in the ternary MXene material using DFT. Although being undoubtedly interesting to a broad community of material scientists, I believe a major revision of the manuscript is required before this work could be reconsidered for publication. The list of comments and suggestions is provided below.

Major points:

1.      I would recommend to revise and strengthen the novelty of the proposed methodology. I would suggest adding in a discussion about how this strategy of combining diffraction studies and simulations can be generalized and applied in solving the structure of other MXene solid solutions.

2.      Since one of the key messages of this study is the solution and refinement of TiNbC structure I would suggest including the corresponding CIF as supporting data.

Minor points:

3.      Could the authors please include the details of how the TiNbC MXene samples were purified for the analysis (XRD, TEM, SEM) or whether they were characterized as received.

4.      It is not clear what is the parent phase of TiNbC MXene. Would it be a Nb substituted Ti3AlC2? Or is the Ti3AlC2 MAX phase first converted into MXene and then doped with Nb? To simulate the crystal structure of TiNbC MXene the authors used the conventional Ti3AlC2 MAX phase as a reference. Would it be possible to simulate the (Ti,Nb)3AlC2 parent phase, once the cation distribution is established for the MXene counterpart? This is to quantify the amount of not fully removed MAX component that is seen in the EDS maps of TiNbC MXenes.

5.      Since the lattice parameters of the investigated TiNbC MXene have been found, I could recommend performing the Le Bail fitting of the experimental diffraction data for the structure refinement.

6.      I would ask the authors to add more specific information when discussing their data on page 3, lines 109-110, “the electron diffraction pattern shows bright spots related to TiNbC nanocrystals with a certain orientation of atomic planes”: is the sample single crystalline or polycrystalline, what distances are observed and what sets of planes they correspond to; page 4, lines 128-130, “The obtained X-ray diffraction pattern 2 in Figure S1 shows that the elimination of Al is accompanied with a change in relative intensities of most of the reflections that is related to the disappearance of certain planes diffracting X-rays”: would it be possible to clarify what are these planes and how the overall symmetry changes upon transformation from MAX to MXene.

7.      Figure 3 caption: is this Ti3C2 MXene or (Ti,Nb)3AlC2 MXene studied in this work?

8.      Figure S3 caption: right hand side panel looks like XRD, and not XRF or reflected electrons.

Author Response

Q1.      I would recommend to revise and strengthen the novelty of the proposed methodology. I would suggest adding in a discussion about how this strategy of combining diffraction studies and simulations can be generalized and applied in solving the structure of other MXene solid solutions.

A1.         Previously, this approach was not used to refine the elementary cell. And this approach claims to be a universal method for determining the positions of atoms in a unit cell for any MXenes.

Q2.      Since one of the key messages of this study is the solution and refinement of TiNbC structure I would suggest including the corresponding CIF as supporting data.

A2.         CIF-file will be added

Q3.      Could the authors please include the details of how the TiNbC MXene samples were purified for the analysis (XRD, TEM, SEM) or whether they were characterized as received.

A3.         The material was characterized in the form in which it was obtained.

Q4.      It is not clear what is the parent phase of TiNbC MXene. Would it be a Nb substituted Ti3AlC2? Or is the Ti3AlC2 MAX phase first converted into MXene and then doped with Nb? To simulate the crystal structure of TiNbC MXene the authors used the conventional Ti3AlC2 MAX phase as a reference. Would it be possible to simulate the (Ti,Nb)3AlC2 parent phase, once the cation distribution is established for the MXene counterpart? This is to quantify the amount of not fully removed MAX component that is seen in the EDS maps of TiNbC MXenes.

A4.         Explanation added: "TiNbC MXene was obtained by etching Al from the MAX phase of (Ti,Nb)3AlC2 with a mixture of HCl and LiF acids. The samples were synthesized by Advanced 2D Materials Co. Ltd. store, China".

Q5.      Since the lattice parameters of the investigated TiNbC MXene have been found, I could recommend performing the Le Bail fitting of the experimental diffraction data for the structure refinement.

A5.         Added the corresponding chart, as well as explanations for the text of the article

Q6.      I would ask the authors to add more specific information when discussing their data on page 3, lines 109-110, “the electron diffraction pattern shows bright spots related to TiNbC nanocrystals with a certain orientation of atomic planes”: is the sample single crystalline or polycrystalline, what distances are observed and what sets of planes they correspond to; page 4, lines 128-130, “The obtained X-ray diffraction pattern 2 in Figure S1 shows that the elimination of Al is accompanied with a change in relative intensities of most of the reflections that is related to the disappearance of certain planes diffracting X-rays”: would it be possible to clarify what are these planes and how the overall symmetry changes upon transformation from MAX to MXene.

A6.         The corresponding analysis was carried out and 2 fragments were added:

1) Regarding TEM data: "The presence of a dot diffraction pattern against the background of an annular diffraction pattern, most likely indicates that a significant part of the nanocrystals of the studied MXene has the same spatial orientation";

2) Regarding XRD data: “This is especially noticeable in the example of plane 1 0 4 (compare the relative intensities of curves 1 and 2 in Figure S1 for a region of about 40 degrees). Exactly in this plane that the excluded aluminum atoms are located, which can be clearly seen on Figure S2a".

Q7.      Figure 3 caption: is this Ti3C2 MXene or (Ti,Nb)3AlC2 MXene studied in this work?

A7.         The title of the figure has been corrected to “X-ray diffraction pattern of the studied of the TiNbC sample”

Q8.      Figure S3 caption: right hand side panel looks like XRD, and not XRF or reflected electrons.

A8.         Figure title corrected

Reviewer 2 Report

This manuscript presents a combination of theoretical and experimental study on the crystal structure of TiNbC MXene. XRD, TEM and EDX characterizations are used to determine the structure and composition of TiNbC MXene. Structural analysis is conducted by comparing the crystal structure of Ti3AlC2 MAX phase with TiNbC MXene to extract the proposed crystal structure of TiNbC MXene. The result is further confirmed by DFT calculations. Overall, I think this manuscript shows considerable impact by determining the crystal structure of TiNbC MXene. However, the following comments should be addressed by the authors before the manuscript can be considered for acceptance:

1. The use of the word "algorithm" in the Introduction section (line 60-66) is misleading. The manuscript demonstrates the analysis approach of comparing crystal structures of the MAX phase with MXenes, but all of the analysis are conducted manually instead of from an algorithm. I would recommend the authors to revise the section accordingly.

2. The authors should specify the sample preparation process for SEM and TEM measurement, because the use of different substrates would have significant impact on the EDX results. Generally, the substrates should not contain the elements of interest in the sample, as in Ti, Nb and C. For SEM/EDX results in Figure 1, the authors should specify the EDX measurement region and show the EDX spectrum. For TEM/EDX result in Figure 2, the authors should label all of the peaks shown in the EDX spectrum and specify the source of these element, either from substrate, sample preparation process or instruments.

3. This manuscript uses several expressions refer the studied MXene, including TiNbC, Ti3-xNbxC2, Ti2Nb4C4, Ti2Nb4C2 (the last one seems to be a typo), which is difficult to keep track of. I would recommend the authors to use only one or two expressions and keep consistent throughout the manuscript. 

4. The explanation on the difference in simulated and experimental XRD patterns of the MXenes in Figure 5 is insufficient. Why the Nb atoms have to stay in the planes with the high XRD intensisty? Even so, the placement of the lattice planes can be shifted. How do the authors determine which atoms to be substituted? After the DFT calculation of the structure with lowest energy, why don't the authors compare the simulated XRD of the calculated structure with the experimental data as well? 

Author Response

Q1. The use of the word "algorithm" in the Introduction section (line 60-66) is misleading. The manuscript demonstrates the analysis approach of comparing crystal structures of the MAX phase with MXenes, but all of the analysis are conducted manually instead of from an algorithm. I would recommend the authors to revise the section accordingly.

A1.         The specified section was modified, the term "algorithm" was excluded from the article

Q2. The authors should specify the sample preparation process for SEM and TEM measurement, because the use of different substrates would have significant impact on the EDX results. Generally, the substrates should not contain the elements of interest in the sample, as in Ti, Nb and C. For SEM/EDX results in Figure 1, the authors should specify the EDX measurement region and show the EDX spectrum. For TEM/EDX result in Figure 2, the authors should label all of the peaks shown in the EDX spectrum and specify the source of these element, either from substrate, sample preparation process or instruments.

A2.         Clarification added: For TEM measurements, MXenes TiNbC samples were mixed in isopropanol and applied to a TED PELLA, Inc. brand grid. Support films Lacey Carbon, 200 mesh, Cu until completely dry. TEM/EDX studies were carried out using an Aztech X-Max 100 energy dispersive analysis attachment. Figure 2,c is corrected.

Clarification added: The TEM-EDX spectrum (Figure 2c) shows peaks corresponding to the MXene material (Ti, Nb, C), O (due to surface oxidation), F, Cl (due to etching of the MAX phase in a mixture of HCl and LiF), Al (remains after etching of the MAX phase), and Cu (due to the copper grid was used)

Q3. This manuscript uses several expressions refer the studied MXene, including TiNbC, Ti3-xNbxC2, Ti2Nb4C4, Ti2Nb4C2 (the last one seems to be a typo), which is difficult to keep track of. I would recommend the authors to use only one or two expressions and keep consistent throughout the manuscript.

A3.         Of course, Ti2Nb4C2 is a typo, it has been corrected. Also, the name of the objects of study, left only 2 options

Q4. The explanation on the difference in simulated and experimental XRD patterns of the MXenes in Figure 5 is insufficient. Why the Nb atoms have to stay in the planes with the high XRD intensity? Even so, the placement of the lattice planes can be shifted. How do the authors determine which atoms to be substituted? After the DFT calculation of the structure with lowest energy, why don't the authors compare the simulated XRD of the calculated structure with the experimental data as well?

A4.         The calculated diffraction pattern after DFT-modeling has been added to the article. It confirms our assumption.

Reviewer 3 Report

In this paper, the authors used both experimental methods (SEM, TEM and XRD) and numerical method (DFT) to determine the composition and crystal lattice parameters of TiNbC MXene. I enjoyed reading the manuscript and recommend it to be published in Materials. However, prior to acceptance, authors should address the following minor comments:

1. In Abstract, “max phase” should be “MAX phase”.

2, In Introduction, “such 24 fields as energy storage, electromagnetic interference shielding, optoelectronics, water de- 25 salination, catalysis, medicine, and many others [2-4].” References should be listed separately for different applications.

3. In Introduction, full name is needed for “XRD”.

4. In Introduction, “MAX phases are fairly new material” should be “MAX phases are fairly new materials”.

5. In Introduction, “In this article we present a new algorithm for refining the structure of 60 MXenes, which have a wide range of applications.” should be “In this article, we present a new algorithm for refining the structure of 60 MXenes, which has a wide range of applications.”

6. In Materials and Method, “Size: 1-5 um” should be clearer, like lateral size.

7. In Materials and Method, “-OH-F-O-Cl” should be “-OH, -F, -O-, -Cl”.

8. In Materials and Method, full name for TEM is needed when it is mentioned for the first time.

9. In Results 3.1., “The corresponding EDX-mapping is presented in Figure 1b. One 97 can see that the main constituting chemical elements Ti, Nb, and C are distributed evenly 98 over the sample surface.” However, C was only observed on substrate. I suppose Figure 1b is the EDX-mapping results for Figure 1a, but no element was shown on the left bottom of the figure.

10. In Results 3.1., areas according to the distance numbers need to be labeled.

11. In Results 3.1., “Figure 2b demonstrates multilayer structure of the MXene corresponding to the 108 SEM image in Figure 1c.” Do you mean the sample is same for SEM and TEM image? If so, how did you do this?

12. In Results 3.1., “The electron diffraction patterns (insets in Figure 2b) show bright 109 spots related to TiNbC nanocrystals with a certain orientation of atomic planes.” Please label the orientation of atomic planes in electron diffraction patterns.

13. In Results 3.2., “Figure 3 shows X-ray X-ray diffraction pattern of the TiNbC sample.” X-ray was repeated.

14. In Supplementary Information, Figure S2: Al toms should be Al atoms.

Author Response

Q1.         In Abstract, “max phase” should be “MAX phase”.

A1.         Corrected

Q2. In Introduction, “such fields as energy storage, electromagnetic interference shielding, optoelectronics, water desalination, catalysis, medicine, and many others [2-4].” References should be listed separately for different applications.

A2.         Corrected

Q3. In Introduction, full name is needed for “XRD”.

A3.         Corrected

Q4. In Introduction, “MAX phases are fairly new material” should be “MAX phases are fairly new materials”.

A4.         Corrected

Q5. In Introduction, “In this article we present a new algorithm for refining the structure of 60 MXenes, which have a wide range of applications.” Should be “In this article, we present a new algorithm for refining the structure of 60 Mxenes, which has a wide range of applications.”

A5.         Corrected

Q6. In Materials and Method, “Size: 1-5 um” should be clearer, like lateral size.

A6.         Corrected

Q7. In Materials and Method, “-OH-F-O-Cl” should be “-OH, -F, -O-, -Cl”.

A7.         Corrected

Q8. In Materials and Method, full name for TEM is needed when it is mentioned for the first time.

A8.         Corrected

Q.9. In Results 3.1., “The corresponding EDX-mapping is presented in Figure 1b. One can see that the main constituting chemical elements Ti, Nb, and C are distributed evenly over the sample surface.” However, C was only observed on substrate. I suppose Figure 1b is the EDX-mapping results for Figure 1a, but no element was shown on the left bottom of the figure.    

A9.The relevant part of section 3.1 is rewritten as follows: «The corresponding EDX-mapping is presented in Figure 1b. One can see that the main constituting chemical elements Ti, Nb, and C are distributed evenly over the sample surface (in the Figure 1b, the carbon signal is visible only on the substrate, which is due to the presence in the sample of a large amount of niobium, which gives a strong signal). In addition to that, a small amount of the parent MAX-phase is visible close to the Al sites. The zoomed fragment of Figure 1a enclosed in the dashed rectangular is depicted in Figure 1c. One can see that the cubic particle is comprised of nanosheets of ~80-120 nm thickness. The EDX-spectrum of the particle shows (Figure 1d) significant amount of Nb - the synthesized compound can be described as Ti1.625Nb3.375C4». Figure 1 has also been corrected to show that the composition of MXene was determined not from mapping, but from the EDX spectrum obtained from a massive particle.

Q10. In Results 3.1., areas according to the distance numbers need to be labeled.

A10.       We do not quite understand the comment of the reviewer. A little more information is needed. Perhaps the numerous changes made to section 3.1 have already taken into account what the reviewer had in mind in this section.

Q11. In Results 3.1., “Figure 2b demonstrates multilayer structure of the Mxene corresponding to the SEM image in Figure 1c.” Do you mean the sample is same for SEM and TEM image? If so, how did you do this?

A11.       The original TiNbC MXene sample was used for both SEM and TEM. But of course they were different parts of MXene.

Q12. In Results 3.1., “The electron diffraction patterns (insets in Figure 2b) show bright 109 spots related to TiNbC nanocrystals with a certain orientation of atomic planes.” Please label the orientation of atomic planes in electron diffraction patterns.

A12.       Unfortunately, atomic planes cannot be noted in this diffraction pattern, because there are no reflections from atomic planes, we see only circular diffraction from many small particles of the sample. But such a diffraction pattern can be used to determine the atomic interplanar distances, which is shown in the left Figure S3.

Q13. In Results 3.2., “Figure 3 shows X-ray X-ray diffraction pattern of the TiNbC sample.” X-ray was repeated.

A13.       Corrected

Q14. In Supplementary Information, Figure S2: Al toms should be Al atoms.

A14.       Corrected

Round 2

Reviewer 1 Report

I would like to thank the authors for addressing all of my comments. I belive the manuscript can be accepted for publication after minor revision:

page 5, line 162: alpha=beta=90o, gamma=120o

Author Response

An unfortunate typo pointed out by respected reviewers has been corrected. Thank you!

Reviewer 2 Report

The authors have addressed the comments from the previous review report. I would like to recommend this manuscript to be accepted by Materials.

Author Response

I thank the dear reviewer for the positive feedback and recommendation for publication